# The Impact of Biotechnologically Produced Lactobionic Acid in the Diet of Lactating Dairy Cows on Their Performance and Quality Traits of Milk

**DOI:** 10.3390/ani13050815

**Published:** 2023-02-23

**Authors:** Diana Ruska, Vitalijs Radenkovs, Karina Juhnevica-Radenkova, Daina Rubene, Inga Ciprovica, Jelena Zagorska

**Affiliations:** 1Faculty of Agriculture, Institute of Animal Sciences, Latvia University of Life Sciences and Technologies, LV-3001 Jelgava, Latvia; 2Division of Smart Technologies, Research Laboratory of Biotechnology, Latvia University of Life Sciences and Technologies, LV-3004 Jelgava, Latvia; 3Processing and Biochemistry Department, Institute of Horticulture, LV-3701 Dobele, Latvia; 4Division of Agronomic Analysis, Research Laboratory of Biotechnology, Latvia University of Life Sciences and Technologies, LV-3002 Jelgava, Latvia; 5Faculty of Food Technology, Latvia University of Life Sciences and Technologies, LV-3004 Jelgava, Latvia

**Keywords:** amino acids, fatty acids, feed, lactating cows, lactobionic acid, milk quality traits, milk yield, whey

## Abstract

**Simple Summary:**

The European food industry creates millions of tons of waste products annually that are discarded or utilized inefficiently. The goals set in European legislation have been pivotal drivers in enhancing waste management and stimulating innovation in recycling. Without innovations in processing technologies, the quantity of waste will steadily rise. Considering the evidence of lactobionic acid’s (Lba) health-promoting benefits and already established protocol for whey lactose conversion via microbial cultures developed by a group from Latvia University of Life Sciences and Technologies LBTU, the current study aimed to elucidate the effect of the supplementation of dairy cows’ diets with biotechnologically obtained Lba-rich whey on animals’ performances and milk quality traits. The acquired results revealed that produced Lba could be deemed an alternative to sugar beet molasses to supplement the diet of dairy cows and positively influence the composition of essential amino acids and polyunsaturated fatty acids. The use of Lba in the diet of dairy cows during the lactation period equal to molasses affected the cows’ performances and milk quality traits, especially fat composition.

**Abstract:**

Dairy processing is one of the most polluting sectors of the food industry as it causes water pollution. Given considerable whey quantities obtained via traditional cheese and curd production methods, manufacturers worldwide are encountering challenges for its rational use. However, with the advancement in biotechnology, the sustainability of whey management can be fostered by applying microbial cultures for the bioconversion of whey components such as lactose to functional molecules. The present work was undertaken to demonstrate the potential utilization of whey for producing a fraction rich in lactobionic acid (Lba), which was further used in the dietary treatment of lactating dairy cows. The analysis utilizing high-performance liquid chromatography with refractive index (HPLC-RID) detection confirmed the abundance of Lba in biotechnologically processed whey, corresponding to 11.3 g L^−1^. The basic diet of two dairy cow groups involving nine animals, Holstein Black and White or Red breeds in each, was supplemented either with 1.0 kg sugar beet molasses (Group A) or 5.0 kg of the liquid fraction containing 56.5 g Lba (Group B). Overall, the use of Lba in the diet of dairy cows during the lactation period equal to molasses affected cows’ performances and quality traits, especially fat composition. The observed values of urea content revealed that animals of Group B and, to a lesser extent, Group A received a sufficient amount of proteins, as the amount of urea in the milk decreased by 21.7% and 35.1%, respectively. After six months of the feeding trial, a significantly higher concentration of essential amino acids (AAs), i.e., isoleucine and valine, was observed in Group B. The percentage increase corresponded to 5.8% and 3.3%, respectively. A similar trend of increase was found for branched-chain AAs, indicating an increase of 2.4% compared with the initial value. Overall, the content of fatty acids (FAs) in milk samples was affected by feeding. Without reference to the decrease in individual FAs, the higher values of monounsaturated FAs (MUFAs) were achieved via the supplementation of lactating cows’ diets with molasses. In contrast, the dietary inclusion of Lba in the diet promoted an increase in saturated FA (SFA) and polyunsaturated FA (PUFA) content in the milk after six months of the feeding trial.

## 1. Introduction

Data published by the EUROSTAT disclose that in 2020 European union farms produced 160.1 million tons (Mt) of raw milk, 1.1% more than in 2019 [1]. Of the total milk obtained, 96.3% was used to produce a range of processed dairy products and fresh products. Among other products, the dairy industry generates a substantial amount of whey which in 2020 corresponded to 55.5 Mt [1]. The relative abundance of water and the high ratio of lactose to protein in dairy whey makes further processing challenging. The report of Ozel et al. [2] reveals that from the total amount of whey generated, only 50% is being processed for the production of high added value products. Whey drying technologies using conventional approaches such as spray dryers are sometimes risky to manufacturers due to extensive fouling and blocking of the production equipment. These challenges increase with higher whey protein levels and temperatures, leading to protein denaturation [3]. Manufacturers are forced to use ultrafiltration and/or reverse osmosis to reduce the risk of equipment clogging along with making whey more solid [4]. However, these approaches challenge small enterprises and are not economically feasible [5], explaining the relatively high price for whey protein isolates. While considering the high biochemical oxygen consumption (BOD) of 50 g L^−1^ and chemical oxygen (COD) of 65 g L^−1^ values, the direct disposal of raw whey is strongly prohibited in many EU countries as it creates serious environmental problems, leading to changes in soil’s physical condition, chemical indicators, and microbiota, thus affecting the yield of crops to be planted [6]. Therefore, it is paramount to find new and cost-effective processing technologies that could stimulate the reuse of whey in many economic sectors, including animal husbandry and food production, and foster a circular economy toward the sustainable development of high added value products from by-products.

The chemical composition of whey is discussed as it varies considerably depending on the milk source and the production process used. However, per 100 g^−1^, it contains an average of 6.5 g of total solids, which includes 5.0 g of lactose, 0.6 g of protein, 0.6 g ash, 0.2 g of non-protein nitrogenous substances, and 0.1 g of fat [7]. Given the composition of whey, especially the high content of lactose, whey represents interest to biotechnologists as a potential source of carbon-containing molecules suitable for being used as nutrients for microbiological cultures. Cutting-edge biotechnology research came with the discovery of functional compounds derived from dairy whey with antioxidant, antimicrobial, antiaging, and immunomodulation activities, such as α-lactalbumin and β-lactoglobulin [8], glycomacropeptide [9], and lactoferrin [10]. Lactobionic acid (Lba) is another functional product that, for the first time, was synthesized chemically by Fischer and Meyer by oxidizing lactose with bromine [11]. To date, the production of Lba has been accomplished via enzymatic biocatalytic [12], electrocatalytic [13], or heterogeneous [14] oxidation and is widely used in some pharmaceutical products as an excipient agent [15]. More recently, a microbially synthesized Lba was obtained under optimized fermentation conditions of cheese whey with *Pseudomonas fragi* [16]. Meanwhile, the ability of *P*. *taetrolens* to produce enzymes involved in the oxidation of acid whey lactose to Lba was highlighted by Sarenkova et al. [17].

Due to intrinsic properties, e.g., the calcium delivery vehicle, acidity regulator, and free radical chelating agent, Lba may represent interest to stockbreeders. Recently, our group achieved encouraging results on the elaboration of Lba from acid whey through a biotechnological approach via lactose oxidation enzymes produced by *P. taetrolens* [18]. Furthermore, the potential utilization of synthesized, isolated, and purified Lba has been demonstrated by Zagorska et al. [19], indicating the ability of Lba to contribute to pig growth performance and enhance the nutritional value of meat proteins. Furthermore, the ability of fermented acid whey permeate to act as a prebiotic while positively affecting the growth and development of normal intestinal microflora of lactating cows was highlighted in an in vivo study performed by Lakstina et al. [20]. Moreover, the inclusion of Lba in laying hens’ diets contributed to the eggshell thickness and their strengthening, as reported in the patent application [21].

These observations, along with limited information regarding the influence of Lba on the productivity of lactating cows and the quality traits of milk, have promoted the design of this study which focuses on the evaluation of the influence of biotechnologically obtained Lba used as a feed supplement in the diet of lactating cows on their performance and the quality traits of milk.

## 2. Materials and Methods

### 2.1. Experimental Design

The study was conducted at the Farm Ruki (Latvia, Vidzeme) and lasted from November 2020 to April 2021 (six months). Two groups of lactating dairy cows were included, i.e., control (Group A) and experimental (Group B). Each group included in the current study was composed of nine animals of two breeds, i.e., Holstein Black and White and Red dairy cows. The animals were up to 60 days in lactation (DIM), representing different lactations (from 1 to 8) divided proportionally between the groups. The basic feed for both groups was prepared directly on the farm as a partial mixed ration (PMR). The composition for one animal included grass silage 41 kg, hay 0.5 kg, rapeseed cakes 3.6 kg, grain flour 8.3 kg, and premix 0.6 kg (JOSERA Cami 0.25 kg, sodium bicarbonate 0.15 kg, DairyPilotFlavoVital^®^ 0.1 kg, lime 0.1 kg, and salt 0.02 kg). Group A was additionally fed 1.0 kg sugar beet molasses, while the diet of Group B was supplemented with 5.0 kg Lba-rich whey. The amount of Lba supplemented into the basic diet was estimated based on the carbohydrate content of molasses. The proximate composition of the liquid fraction rich in Lba and molasses is given in Table 1. Dairy cows were housed under tie stalls and individually fed and watered ad libitum. Cows were milked twice daily.

### 2.2. Dairy Lactating Cows’ Performances and Quality Traits of Milk Samples

Milk productivity traits: milk yield kg d^−1^ and sampling for testing were conducted twice per month during the experiment. Milk composition and quality indices were determined at the beginning and end of the experiment. Raw milk samples were taken during morning milking and divided into two parts: one part was preserved using Broad Spectrum MicroTabs II (BSM II) and immediately delivered to the Dairy Laboratory, Ltd. The analysis of raw milk quality traits, i.e., fat, protein, casein, and urea content, was performed using the MilkoScan FT6000 (FOSS, Hilleroed, Denmark) mid-infrared spectroscopic approach following the guidelines outlined in the ISO 9622|IDF 141:2013. The estimation of somatic cells was performed using an instrumental flow cytometry method by Fossomatic™ (FOSS, Hilleroed, Denmark) according to the ISO standard 13366-2|IDF 148-2:2006. The second part of the raw milk samples devoted to the analysis of fatty acids (FAs) and amino acids (AAs) was kept at a temperature of −20 ± 1 °C until further processing and analysis, a maximum of two weeks.

The somatic cell count (SCC) per 1 mL of milk was converted to standardized units, i.e., somatic cell score (SCS), by using the following equation [22]:(1)$$SCS={log}_{2}\left(\frac{SCC}{100}\right)+3$$
where SCC is somatic cells per mL of milk.

The SCS in milk was used as a quality and indirect animal health indicator.

In order to evaluate results among groups and study phases (beginning and end of the experiment), the milk yield and its composition were transformed to energy-corrected milk (ECM), which indicates the amount of energy in the milk considering the values of milk, fat, and protein yield (ICAR, 2017). The ECM was determined following the equation
(2)$$ECM=\frac{FY\times 38.3+PY\times 24.2+MY\times 0.7832}{3.14}$$
and was expressed in kg d^−1^, where ECM is energy-corrected milk, FY is fat yield in kg, PY is protein yield in kg, and MY is milk yield in kg.

The predicted milk protein efficiency ratio (PER) was calculated according to three equations proposed by Alsmeyer, Cunningham, and Happich (1974), taking into account the values of AAs:(3)$$PER1=-0.684+0.456\left(LEU\right)-0.047\left(PRO\right)$$
where LEU and PRO are the content of leucine and proline, respectively.
(4)$$PER2=-0.468+0.454\left(LEU\right)-0.105\left(TYR\right)$$
where LEU and TYR are the content of leucine and tyrosine, respectively.
(5)$$PER3=-1.816+0.435\left(MET\right)+0.780\left(LEU\right)+0.211\left(HIS\right)-0.944\left(TYR\right)$$
where MET, LEU, HIS, and TYR is the content of methionine, leucine, histidine, and tyrosine, respectively.

The ratio of essential AAs (E) to the total AAs (T) of the protein was calculated based on the equation provided by Chavan et al. [23]:(6)ET=∑EAA∑TAA×100%
where E/T is a ratio of essential amino acids (E) to the total amino acids (T); ∑_EAA_ is the sum of essential amino acids; and ∑_EAA_ is the sum of total amino acids.

The index of atherogenicity (IA) which characterizes the atherogenic potential of fatty acids, the index of thrombogenicity (IT), a ratio of hypocholesterolemic to hypercholesterolemic (HH) values, and the health-promoting index (HPI) were calculated according to Equations (7–10) proposed by Chen and Liu (2020).
(7)$$IA=\frac{C12:0+\left(4\times C14:0\right)+C16:0}{{}_{\sum UFA}}$$
where IA is the index of atherogenicity and ∑_UFA_ is the sum of unsaturated fatty acids.
(8)$$IT=\frac{C14:0+C16:0+C18:0}{\left(0.5\times \sum_{MUFA}\right)+\left(0.5\times \sum_{n-6PUFA}\right)+\left(3\times \sum_{n-6PUFA}\right)+\sum \frac{n-3}{n-6}}$$
where IT is the index of thrombogenicity; ∑_MUFA_ is the sum of monounsaturated fatty acids; and ∑_PUFA_ is the sum of polyunsaturated fatty acids.
(9)$$HH=\frac{cis-C18:1+\sum_{PUFA}}{C12:0+C14:0+C16:0}$$
where HH is a ratio of hypocholesterolemic to hypercholesterolemic values; *cis* implies the isomeric form of C18:1 fatty acid; and ∑_PUFA_ is the sum of polyunsaturated fatty acids.
(10)$$HPI=\frac{\sum_{UFA}}{C12:0+\left(4\times C14:0\right)+C16:0}$$
where HPI is the health-promoting index and ∑_UFA_ is the sum of unsaturated fatty acids.

### 2.3. Chemicals, Standards, and Reagents

A mixture of C_4_-C_24_ fatty acid methyl esters (FAMEs) and amino acids (AAs) with a purity of ≥99.0% were acquired from Sigma-Aldrich Chemie Ltd. (St. Louis, MO, USA). Acetonitrile, methanol, *n*-hexane, and formic acid (puriss p.a., ≥98.0%) of liquid chromatography–mass spectrometry (LC-MS) grade were purchased from the same producer. Lactobionic acid (Lba) with purity ≥97.0% was obtained from Acros Organics (Geel, Belgium). Ammonium hydroxide solution (25% *v*/*v*) and diethyl ether (puriss p.a., ≥99.5%) were obtained from Chempur (Piekary Śląskie, Silesia, Poland). Hydrochloric acid (37% *v*/*v*) was purchased from VWR™ International, GmbH (Darmstadt, Germany). Sodium hydroxide, potassium hydroxide, phenolphthalein, and 0.5 M trimethylphenylammonium hydroxide solution (TMPAH) in methanol for GC derivatization were of reagent grade and were obtained from Sigma-Aldrich Chemie Ltd.

### 2.4. Production of Lactobionic Acid from Pre-Concentrated Whey

In this study, pre-concentrated whey obtained from a local producer Jaunpils Ltd. (Jelgava, Latvia), with a proximate composition depicted in Table 2, was used as a carbon source for *P*. *taetrolens* DSM 21104 during the production of Lba. The description of the operational and process conditions for the production of Lba was provided in detail in our previous study [26].

### 2.5. The HPLC-RID-DAD Analytical Conditions for Lactobionic Acid and Lactose Determination

Lba was analyzed using a Shimadzu series LC-20 high-performance liquid chromatography system equipped with the SPD-M20A photodiode-array detector (Shimadzu Corporation, Tokyo, Japan). All samples before HPLC analyses were centrifuged at 14,200× *g* for 10 min to remove cell debris and other water-insoluble substances. The LBA was determined using a refractive index detector RID-10A (Shimadzu Corporation, Tokyo, Japan). Chromatographic separation of Lba was carried out using a hybrid silica-based YMC-C18 column (4.6 mm × 250 mm, 5 µm; YMC, Kyoto, Japan) operating at 40 °C and a flow rate of 1.0 mL min^−1^. The separation of Lba was conducted using an isocratic mobile phase with 2 L elution containing 1.15 mL H_3_PO_4_, 14.36 g KH_2_PO_4_, and 20 mL acetonitrile. The detection wavelength was set at 210 nm. The injection volume was 10 μL.

The quantitative analysis of lactose was performed using the same system while utilizing a refractive index detector RID-10A (Shimadzu Corporation, Tokyo, Japan). Chromatographic separation was conducted using an Altima Amino (4.6 × 250 mm; 5 μm; Grace™, Columbia, MD, USA) column. The temperature of the column and flow cell was maintained at 30 °C. A mixture of H_2_O and MeCN (75:25, *v*/*v*) was used as the mobile phase in the isocratic mode. The flow rate of the mobile phase was 1.0 mL min^−1^. The injection volume was 10 μL. System control, data acquisition, analysis, and processing were performed using Empower 3 Chromatography Data Software version (build 3471) (Waters Corporation, Milford, MA, USA).

### 2.6. Acid Hydrolysis of Milk for Amino Acid Determination

Before acid hydrolysis, each milk sample was defatted and freeze-dried to obtain protein isolates. Then, the prepared isolates 200.0 mg ± 0.1 were subjected to acid hydrolysis with 5.0 mL of 6M HCl solution according to the ISO 13903:2005 standard with modifications. The hydrolysis was undertaken in 22.0 mL glass Headspace chromatography bottles (PerkinElmer, Inc., Waltham, Massachusetts, USA) with screw caps and silicone seals in a drying cabinet of Pol-Eko Aparatura SP.J. (Wodzislava Slonska, Poland) at a temperature of 110 °C for 24 h. Before hydrolysis, to slow down the oxidation–reduction reaction of the compounds of interest, the stabilizing reagent phenol was added directly to the sample in the amount of 0.02% (*w*/*w*). After hydrolysis, the volume of hydrolysate was adjusted to 7.0 mL with H_2_O, and it was normalized to 6.5–6.8 using 2.18 mL of 25% NH_4_OH solution. The final volume was 10.0 mL. The obtained hydrolysate was subjected to 1 min intensive Vortexing using the “ZX3” vortex mixer (Velp^®^ Scientifica, Usmate Velate, Italy), followed by centrifugation at 16,070× *g* for 10 min at 19.0 ± 1 °C in a “Hermle Z 36 HK” centrifuge (Hermle Labortechnik, GmbH, Wehingen, Germany). Before LC-MS analysis, the collected supernatant was filtered using a 0.22 µm hydrophilized polytetrafluoroethylene (H-PTFE) membrane filter (Macherey-Nagel GmbH & Co. KG, Dueren, Germany).

### 2.7. The HPLC-ESI-TQ-MS/MS Analytical Conditions for Amino Acids

The chromatography analysis of AA was conducted using a “Shimadzu Nexera UC” series liquid chromatography (LC) system (Shimadzu Corporation, Tokyo, Japan) coupled to a triple quadrupole mass-selective detector (TQ-MS-8050, Shimadzu Corporation, Tokyo, Japan) with an electrospray ionization interface (ESI). A sample of 3 µL was injected onto a reversed-phase “Discovery^®^ HS F5-3” column (3.0 µm, 150 × 2.1 mm, Merck KGaA, Darmstadt, Germany) operating at 40 °C with a flow rate of 0.25 mL min^−1^. The mobile phases used were acidified H_2_O (1.0% HCOOH *v*/*v*) (A) and acidified MeCN (1.0% HCOOH *v*/*v*) (B). The program of stepwise gradient elution of the mobile phase B for 20 min was implemented as follows: T_0 min_ = 5.0%, T_5_._0 min_ = 30.0%, T_11_._0 min_ = 60.0%, T_12_._0 min_ = 80.0%, and T_12_._1 min_ = 5.0%. Finally, re-equilibration for 3 min was conducted after each analysis following the initial gradient conditions. The MeCN injections were included as a blank run after each sample to avoid the carry-over effect. Data were acquired using “LabSolutions Insight LC-MS” version 3.7 SP3, which was also used for instrument control and processing. Ionization in the positive ion polarity mode was applied in this study. At the same time, data were collected in profile and centroid modes, with a data storage threshold of 5000 absorbance for MS. The operating conditions were as follows: detector voltage 1.98 kV, conversion dynode voltage 10.0 kV, interface voltage 4.0 kV, interface temperature 300 °C, desolvation line temperature 250 °C, heat block temperature 400 °C, nebulizing gas argon (Ar, purity 99.9%) at a flow rate of 3.0 L min^−1^, heating gas carbon dioxide (CO_2_, purity 99.0%) set low at 10.0 L min^−1^, and drying gas nitrogen (N_2_, separated from air using a nitrogen generator system from “Peak Scientific Instruments Ltd.” (Inchinnan, Scotland, UK), purity 99.0%) at flow 10.0 L min^−1^. All AAs were observed in the programmed and optimized multiple reaction monitoring (MRM) mode. Quantitative analysis of AAs was performed by injecting 3.0 μL of calibration solution at 15 °C with the range of 0.075–2.5 μM L^−1^. The working solution was prepared immediately before being used. Representative chromatographic separation of 18 AAs is given in Figure 1.

### 2.8. Preparation of Lipid Fraction via Alkaline-Assisted Hydrolysis with Subsequent Liquid–Liquid Extraction

The isolation of lipophilic fraction from dry milk cream (please refer to Section 2.6*. Acid Hydrolysis of Milk for Amino Acid Determination*) was performed following the procedure described by Radenkovs et al. [27] with minor modifications. For the release of bound forms of fatty acids (FAs), 10% (*w*/*v*) KOH dissolved in 80% MeOH (MeOH:H_2_O ratio 80:20 *v/v*) was applied. Briefly, duplicate samples of 3.0 ± 0.1 g of freeze-dried cream obtained after milk separation were weighed in 50 mL reagent bottles with screw caps. For the hydrolysis of FAs, 30 mL of freshly prepared methanolic KOH was added to each cream sample, and the mixture underwent incubation in a water bath “TW8” (Julabo^®^, Saalbach-Hinterglemm, Germany) at 65 °C temperature for 3 h. After hydrolysis, the cleavage of bonds present in the potassium salts of FAs was obtained by adjusting the pH of the solution to pH 2.0 ± 0.2 by adding 3.3 mL HCl (6M). The extraction of FAs was implemented via liquid–liquid phase separation using *n*-hexane as the sole solvent. After hydrolysis, samples were cooled to room temperature (22 ± 1 °C) and quantitatively transferred to Falcon 50 mL conical centrifuge tubes (Sarstedt AG & Co. KG, Nümbrecht, Germany). Afterwards, 10 mL of *n*-hexane was added to each tube, followed by vortex-mixing for 1 min. Finally, the layers were separated via centrifugation at 3169× *g* for 10 min in a “Sigma, 2-16KC” centrifuge (Osterode near Harz, Germany). The top *n*-hexane layer was decanted and collected. The extraction procedure was repeated three times. First, the resulting lipid fraction (30 mL) was evaporated using a “Laborota 4002” rotary evaporator (Heidolph, Swabia, Germany) at 65 °C, and the dry residues were then reconstituted in 5 mL of *n*-hexane and filtered through a polytetrafluoroethylene hydrophobic (PTFE) membrane filter with a pore size of 0.45 µm. The filtrates were quantitatively transferred to 20 mL scintillation glass vials and subjected to drying under a gentle stream of N_2_ to complete dryness. Prepared samples were kept at a temperature of −18 ± 1 °C until further analysis and were used within a maximum of two weeks. Before GC-MS analysis, obtained dry lipid fractions were reconstituted in 2 mL pyridine.

### 2.9. Preparation of Fatty Acids for GC-MS Analysis

The TMPAH reagent was used as a methylation agent of the functional groups to obtain volatile FAMEs derivatives. The methylation procedure was performed following the methodology ensured by Radenkovs et al. [27].

### 2.10. The GC Conditions for FAMEs Analysis

The analysis of FAMEs was performed using a “Clarus 600” system PerkinElmer, Inc. (Waltham, MA, USA) coupled with a quadrupole “Clarus 600 C” mass-selective detector (Waltham, MA, USA). The conditions were adopted from Radenkovs et al. [27].

### 2.11. Statistical Analysis

The obtained data were analyzed using descriptive statistics, and the differences between the study groups and phases of the study were assessed using ANOVA with Student’s *t*-test correction, setting the confidence level at *p* ≤ 0.05. Statistical processing of the data was carried out using the MS Office program Excel version 2016 (Microsoft Corporation, Redmond, Washington, USA).

## 3. Results and Discussion

### 3.1. Animals’ Performances and Quality Traits of Milk

Balanced feeding for lactating cows, especially at the beginning of lactation, is crucial [28] as it influences cows’ performances, overall health, and milk quality traits. The selection of a supplement to compensate for the lack of energy in feed depends on factors such as feeding technology, supplement availability in the market, and costs. In the current study, sugar beet molasses for Group A and Lba for Group B were selected as supplements to compare the effect on cows’ performances and milk quality traits. The productivity results were analyzed for each group at the study’s beginning and end (see Table 3). As seen at the beginning of the experiment, the milk yield values were not significantly different (*p* ≥ 0.05) between the study groups. However, a substantial difference (*p* ≤ 0.05) was observed comparing milk yield within the study phase between initial and final values. It was observed that at the end of the experiment passing six months, the milk yield in Group A decreased by 19.3%, while in Group B, the decrease amounted to 23.0%, which was 3.7% higher in Group A (Table 3). No apparent influence of dietary treatment on lactation performance was found; this observation is in line with Penner and Oba [28]. The decrease in milk yield is explained by the lactation phase, which directly influences the milk yield as the number of lactation days increased during the study [29,30]. A similar observation was made by Vijayakumar et al. [31], indicating that cows with the second lactation produced 24.18% greater milk than the first lactation cows, while the fourth lactation cows showed a decreased milk yield by 16.04% from the third lactation. According to a study reported by the National Research Council [32], fat content in milk is the most varying value, while lactose is the least, and this observation was also reinforced in this study. As seen, the fat content between the groups within the beginning phase of the experiment varied significantly (*p* ≤ 0.05), corresponding to 23.5% (Table 3). The percentage difference between the groups at the end of the experiment amounted to 15.4%, which was 8.1% lower than at the beginning. Such a difference could be the case of the animals’ physiological states, e.g., the availability of hormones such as adrenaline and noradrenaline that are reported to be responsible for lipolytic activity in adipose tissue [33]. At the end of the experiment, the most apparent increase in fat content was found in Group A, corresponding to a 14.7% increase compared with the initial value. A similar increase in fat content was also observed in Group B, though this value corresponded to 7.1%. It was reported that the supplementation of molasses in dairy cows’ diets substantially contributes to a higher fat yield and concentration of fatty acids in primiparous cows [34]. The increase in fat content of Group A fed with a basic diet supplemented with molasses can be explained by an enhanced rumen fermentation process moderated by pH, thus promoting mammary de novo fatty acid synthesis [35,36]. This statement was further reinforced by [37], indicating the increase in effective ruminal degradability (ERD) of dry matter in an in situ ruminal study. It is worth noting that the supplementation of lactating cows’ diets with Lba could be considered a promising carbon-containing alternative to molasses, ensuring an increase rather than a decrease in fat in milk without affecting acidosis.

Multiple pieces of scientific evidence have revealed that milk composition, especially crude protein and fat content, strongly depends on the milk yield [38]. Hence, highly productive dairy cows will provide a lower protein yield in milk than those with low productivity [39]. However, one of the critical factors determining the amount of protein in milk is the availability of nutrients, especially those rich in proteins, that the animal ingests in the feed [40]. The results of this study imply that the content of protein in milk from the dairy cows of Group A who received a high feed diet rich both in carbohydrates and protein (Table 1) was found to be significantly (*p* ≤ 0.05) higher compared with the initial value, corresponding to a 9.1% increase.

Previous studies also observed an increase in protein content, reflecting a higher nutritional value of milk from low-productivity dairy cows [41]. However, in Group B, in which the feed of animals was supplemented with biotechnologically produced Lba, the protein content, considering the lower milk yield at the end of the experiment, remained intact, corresponding to 3.8%. The observed values are consistent with those reported by Murphy [42]. A plausible explanation for obtaining higher values of protein in milk from Group A relies upon the availability of readily digestible compounds present in molasses, such as sucrose, fructose, and glucose [24,40], while in Lba-rich whey solution, the main representative is lactose.

Casein and whey protein are milk’s major proteins, and casein corresponds to roughly 80% of the total protein in bovine milk [43]. The initial values of casein in milk samples fluctuated from 2.6 to 3.0%, with Group B having the highest content and Group A having the lowest (Table 3). The observed values are consistent with those of Guo and Wang [44]. As with protein, the content of casein was affected by feeding. The highest content was found in Group A at the end of the experiment, corresponding to an increase of 7.7%. In turn, no changes were observed in Group B after six months of the feeding trial. The results indicate that optimizing feed intake with ingredients rich in carbohydrates and organic acids such as molasses or Lba can ensure the necessary energy level to retain milk’s nutritional value (casein in particular) during lactating. This observation is in line with those proposed by Emery as far back as four decades ago, in 1978 [45].

The SCC is a direct marker of mastitis infection for individual cows and within herds and therefore was evaluated critically as an indirect indicator of cow udder health [46]. To better reflect the state of animal health, the SCS values were calculated in this study and are depicted in Table 3. According to Shook and Schutz [47], having these numbers allows for achieving genetic improvement and better results in controlling mastitis resistance. As seen, the initial values of SCS varied from 3.2 and 2.3, with Group A at the initial stage of the experiment having the highest value and Group B having the lowest, respectively (Table 3). The estimation made available by Smith et al. [48] was that cows with an SCS of 0–3 are generally considered to be uninfected. At the end of the experiment, the SCS values varied in the range from 3.0 to 3.5, with Group A showing the highest value while Group B showed the lowest. The most apparent increase in SCS values was found in Group B. Although, such an increase in SCS only marginally contributed to the reduction in milk yield, as proposed by Smith et al. [48].

It has been proposed that the content of urea in milk can be utilized as a non-invasive input to a system to monitor the crude protein status in dairy cows on a regular basis [49]. Therefore, urea content in milk samples was used in this study as a biomarker to estimate the availability of AAs in the diet of animals. The initial values of urea content varied from 23.3 to 23.5 mg dL^−1^. The observed values were consistent with those reported by Rzewuska and Strabel [50] for the dairy cow in the first phases of lactation (Table 3). However, after six months of the experiment, following the developed dietary treatment schedule, the amount of urea in the milk of Group A and Group B decreased by 35.2% and 21.7%, respectively. Nevertheless, the observed values comply with data reported by Duinkerken et al., 2011 [51], indicating the range of urea in milk from 15.0 to 30.0 mg dL^–1^ as being optimal. Since the composition of molasses is mainly represented by carbohydrates while lacking essential AAs such as methionine, histidine, and lysine [52], a relatively higher value of urea content in Group B can be explained. Incorporating biotechnologically produced Lba into the diet of dairy cows resulted in ensuring the availability of readily digestible AAs essential for animals [53,54].

Since the ECM is a generic productivity indicator that provides a clue on the value of milk based on the milk yield, fat, and protein content, this value is widely used to assess the overall quality of obtained milk as a function of dietary treatment [39]. As seen, the ECM values at the initial stage of the experiment were significantly different (*p* ≤ 0.05) between the study groups (Table 3). The observed values were considerably higher than those reported by Guinguina et al., 2020, [55] for dairy cows with a basic-feed diet while they were significantly lower than for dairy cows fed rumen-protected lysine as a supplement to the basic diet [56]. The ECM values changed significantly (*p* ≤ 0.05) at the end of the study, corresponding to a percentage reduction of 11.4% and 23.2% for Group A and Group B, respectively. The main factor contributing to the decrease in ECM values was milk yield, which dropped the most in Group B fed with Lba. The report of Miller et al., 2021, [34] indicates that molasses with 34% sucrose positively contributed to dairy cows’ performances during the *postpartum* period, along with improved milk quality traits, by stimulating ruminal butyrate production and papillae development. Moreover, Ravelo et al. [57] also concluded that by-products rich in sucrose or lactose in the diet of dairy cows promoted ruminal microbial fermentation, encouraging digestibility and increasing the pH of rumen fluids.

Overall, the use of Lba in the diet of dairy cows during the lactation period favorably affected the performance and quality traits of milk; however, to achieve better results, the optimization of feed intake with ingredients rich in carbohydrates and proteins such as molasses and biotechnologically produced Lba, respectively, can deliver the necessary energy levels for increased milk production and its quality.

### 3.2. The Changes in Amino Acids and Their Quality Indices in Relation to Feeding Trial

Met, Lys, and His have been identified as the most limiting AAs for lactating dairy cows [53], and their lack in the diet of animals leads to limited milk protein, fat production, and milk yield [58]. Therefore, it has been proposed that supplementing the diet of lactating cows with rumen-protected AAs may be a prosperous approach for improving animals’ performances and the quality traits of milk [59].

The AA profile was analyzed via the HPLC-ESI-TQ-MRM-MS/MS approach, by-passing the derivatization step to elucidate the quality of proteins obtained in produced milk. The selective analysis confirmed the presence of 17 AAs in all milk samples except tryptophan due to its high oxidative degradation (Table 4). Furthermore, it was observed that Glu, Leu, Pro, and Lys were the prevalent representatives of AAs in milk protein. The observed values are consistent with those reported by Landi, Ragucci, and Di Maro [60]. In the course of further study of the content of total AAs in milk protein, no significant difference between Group A and Group B was found at the beginning of the experiment. However, statistically significant differences (*p* ≤ 0.05) were established between Group A and Group B in the content of individual AAs such as Ile, Lys, and Val. After six months of the feeding trial, a significantly higher concentration of Ile and Val was detected in Group B, and the percentage increase corresponded to 5.9% and 3.3%, respectively. An increase in Tyr content by 6.5% and 4.3% was also observed in Group A and Group B, corresponding to the value of 4.9 g 100 g^−1^ in both groups. It is worth noting that the most apparent increase in the content of Ala was observed in Group B, indicating that cows responded favorably to Lba rather than to molasses. The study also noted a significant decrease (*p* ≤ 0.05) in Gly content by 10.5% and 5.3% in Group A and Group B, respectively. A similar observation was made by Li et al. 2019 [61], performing the metabolic profiling of yak mammary gland tissues and speculating that the decrease in Gly was related to the negative energy balance in yaks. The most pronounced decrease in Thr content was found in the milk of Group B, corresponding to 4.7%, while in Group A, a reduction amounted to 2.3%. A plausible explanation for having a reduction in Thr has been given by Tang et al., 2021 [62], indicating that the presence of this essential AA in high concentrations is vitally important to newborns since its primary function is to provide antimicrobial activity against pathogenic bacteria and to modulate the immune system response to viruses, while its gradual decrease takes place as the calf grows.

Branched-chain AAs (BCAAs, valine, leucine, and isoleucine) belong to the group of exogenous AAs that must be supplied to the body through the diet [63]. Multiple beneficial effects of BCAAs have repeatedly been proven [64,65], so the importance of these AAs in human nutrition is undebatable. The results of this study revealed a relatively similar sum of BCAAs in the milk sample at the beginning of the experiment with the dietary treatment. The content varied from 20.2 to 20.7 g 100 g^−1^, with Group A having the lowest value and Group B having the highest value. The observed values are consistent with those of Hulmi et al., 2010 [66]. It is worth noting that the content of BCAAs in Group A after six months of the feeding trial decreased by 0.5%, while in Group B it increased by 2.4%.

The predicted protein efficiency ratio (PER) is a valuable method providing crucial information on the quality of proteins in food systems. However, utilizing in vivo models to estimate PER is considered time-consuming and expensive [67]. In this study, we attempted to predict the PER values based on mathematical equations using the information on amino acids from the milk samples. According to these models, the PER1 (Leu and Pro), PER2 (Leu and Tyr), and PER3 (Met, Leu, His, Tyr) values as functions of the AAs selected were estimated and they are depicted in Table 4. The results of the present study showed that PER1 values for protein isolates obtained from Group A and Group B before the feeding trial lay within the range between 3.1 and 3.2, respectively. Based on Friedman’s classification, a PER < 1.5 is to be considered poor, from 1.5 to 2.0 is considered to be moderate, and > 2.0 is considered to be superior [68]. Based on this proposal, the protein isolates can be classified as highly digestible, close to the values reported by Lee et al. [69] for proteins of deboned chicken meat. In addition, Sarwar [70] and Dupont and Tomé [71] support our observation, indicating that nearly 95% of milk protein is readily digestible within in vivo gastrointestinal tract models. It is worth noting that the observed values were far from those indicated for extruded, cooked, and baked yellow and green split pea flour [72]. Since the PER1 values for Group A and Group B were close to each other after a feeding trial of six months, it is believed that animals received a balanced diet and even energy distribution within the entire experimental period.

However, for calculation using Leu and Tyr, relatively higher values were determined for the PER2 index. The observed values after the feeding trial ranged from 3.2 to 3.3 for Group A and Group B, respectively. A relatively high value of Tyr can explain the difference between PER1 and PER2, reported to have roughly 99.0% true ileal digestibility [73]. Group B tended to show a higher value after a feeding trial with Lba than Group A who were fed a high sucrose diet. The higher PER2 values compared to PER1 were mostly Leu-concentration-dependent.

The predicted PER3 values were found to be different from those of PER1 and PER2 due to the inclusion of additional AAs, which were speculated to be more accurate. The assessed values after the feeding trial for both groups were statistically similar, corresponding to 2.7. However, after the feeding trial, the digestibility rate worsened despite the increase in Tyr and Ile, estimated by a percentage reduction of 11.1% and 3.7% for Group A and Group B, respectively. On the other hand, the increase in Phe and Ala concentrations and the rearrangement of AAs were the main factors that did affect the lowering of the digestibility rate of milk proteins. However, a decrease in PER3 values observed by Pastor-Cavada et al. [74] seemed to be Met-content-dependent. According to PER3 values, the predicted digestibility of milk protein isolates above the standardized PER value for casein of 2.5 indicates its high bioavailability with a digestibility rate close to bean proteins, as Mecha et al. reported [75]. Finally, the ratio of essential AAs to the total AAs at the beginning and end of the study was found in the range from 44.9 to 45.5% and from 45.6 to 45.8% for Group A and Group B, respectively.

The observed ratio values of essential to total AA comply with the quality criteria outlined by the FAO/WHO [76]. Given these numbers, it is attainable to state that milk obtained following lactating cows’ dietary treatments should be considered to be an excellent source of amino acids that could provide the body with all essential AAs.

### 3.3. The Changes in Fatty Acids and Their Nutritional Indexes of Milk Lipids in Relation to Feeding Trial

The composition of FAs in lipids recovered from milk cream is depicted in Table 5. In total, 29 FAs were identified and quantified, among which the dominance of palmitic acid (C16:0) from 31.0 to 34.7%, followed by oleic acid (C18:1n9c) from 18.4 to 20.9%, myristic acid (C14:0) from 11.8 to 12.9%, stearic acid (C18:0) from 8.5 to 10.1%, linolelaidic acid (C18:2n6t) from 1.5 to 2.3%, linoleic acid (C18:2n6c) from 1.6 to 1.8%, and behenic acid (C22:0) from 1.4 to 2.1% was found. The results are consistent with those of Månsson [77], indicating a similar descending order of FA content recovered from bovine milk. The results of the present study indicated that a high feed diet rich in sucrose (Group A) negatively affected the amount of individual FAs in the milk. The most apparent decline in the content of major FAs was observed for linoleic acid (C18:2n6t), corresponding to a 21.6% loss. In contrast, no changes for this polyunsaturated FA were found in Group B fed with biotechnologically obtained Lba. Similar changes were observed for behenic acid (C22:0), corresponding to a 21.6% loss and a 21.4% increase for Group A and Group B. A positive influence of biotechnologically obtained Lba was identified for FAs such as lauric (C12:0), tridecanoic (C13:0), myristoleic (C14:1), pentadecanoic (C15:0), and pentadecanoic (C15:1) acids.

The negative effect of molasses supplementation on dairy cows’ performances and milk composition by reducing milk yield, milk protein, lactose yield, and the composition of unsaturated FAs, in particular, was reported by Torres et al. [78]. In some cases, there was a marked decrease in oleic acid after using molasses as an additive to feed lactating cows due to the interaction of molasses with buffers that negatively affected ruminal fermentation and consequently led to a loss in milk quality [79]. However, after animals received a molasses diet, this experiment observed an increase rather than a decrease in oleic acid (C18:1n9c) concentration in milk. The content of CLA is feed-type-dependent since its concentration greatly varies from report to report [80]. However, the CLA values found are in direct agreement with those reported by Brito et al. [81] for cows fed chiefly grass. The CLA concentration in milk varied from 0.6 to 1.0%, with Group A fed with molasses having the highest value and Group B supplied with Lba having the lowest. After six months of the feeding trial, a significant reduction (*p* ≤ 0.05) in CLA was noted in Group A, corresponding to a 30.0% loss. On the other hand, the Lba positively affected CLA in Group B since as much as a 33.3% increase was observed. Relatively higher values of CLA in Group B can be explained presumably by the chemical composition of a liquid fraction rich in Lba, particularly the availability of linoleic acid that promoted the synthesis of CLA as reported by Gómez-Cortés et al. [82].

Overall, the content of FAs in milk samples was affected by feeding. Without reference to the decrease in individual FAs, the higher values of MUFAs were achieved by the supplementation of lactating cows’ diets with molasses (Group A). In contrast, the dietary inclusion of biotechnologically produced Lba in the diet promoted the increase in the content of SFAs and PUFAs in the milk.

In the present study, health-related lipid indexes IA, IT, HH, and HPI were established to better reflect the health-promoting properties of milk lipids (Table 5).

It was highlighted that IA and IT are great tools for assessing the potential contribution of FAs to human health [83]. The lower IA and IT values, the less risk of developing cardiovascular diseases caused by blood vessel clogging. Performing mathematical analysis, the following IA indices were obtained from 2.8 to 3.6 and from 2.4 to 3.4 for Group A and Group B at the beginning and end of the study, respectively. The observed values are consistent with those of Lobos-Ortega et al. [84] for fresh bovine milk estimated using near-infrared spectroscopy. IA values were statistically different (*p* ≤ 0.05) between Group A and Group B at the study’s beginning and end. Up to 28.6% and 41.7% increases in IA values were observed for Group A and Group B, passing six months of the feeding trial, respectively. The lower IA value in Group B is explained by the statistically higher concentrations of individual MUFAs. Additionally, statistically higher CLA values in Group B reinforce this speculation as their superior anti-atherogenic, anti-platelet, and antioxidant properties have been reported multiple times by [85,86].

However, the opposite results were obtained concerning the IT index, indicating statistically lower values for Group A than for Group B, corresponding to 3.0 and 3.5, respectively. To a greater extent, the observed IT values in both groups corresponded to those reported by Silva et al. [87] and Sharifi et al. [88] for milk from crossbred cows subjected to feed ad libitum and from high forage and nitrate-fed Holstein lactating cows, respectively. The enhancement of the rumen fermentation process and pH optimum achieved by supplementing the diet of Group A with readily digestible mono and disaccharides present in molasses perhaps promoted mammary *de novo* FAs rather than those with anti-atherogenic activity synthesis as proposed by Palmquist, Beaulieu, and Barbano [35]. Further analysis revealed no statistically significant differences (*p* ≥ 0.05) between the HH values, indicating equal values for all samples investigated, corresponding to 0.5.

For the first time, the HPI was proposed by Chen et al. [89] as a quality marker of dietary fat, which is presently widely used in the analysis of dairy products. Furthermore, it is believed that products with higher HPI values are supposed to be more beneficial to human health [19]. The HPI values were found in the range from 0.3 to 0.4, and no statistically significant differences (*p* ≥ 0.05) for both groups were revealed at the end of the feeding trial (Table 5). The observed values are in line with those reported by Kasapidou et al. [90] for sheep fed in confinement with no access to grazing and by Chen and Liu [91] for the cream of Holstein cows.

## 4. Conclusions

The abundance of Lba in the whey fraction obtained by taking advantage of optimized fermentation conditions developed by the group of LBTU utilizing *P*. *taetrolens* DSM 21104 was confirmed chromatographically, corresponding to 11.3 ± 0.3 g L^−1^. A substantial yield of functional Lba made it attainable to enrich the diet of lactating cows (Group B) with the functional component, which has been used as an alternative to sugar beet molasses (Group A). The results of this study revealed an equally effective contribution of the Lba supplementation on dairy cow performance compared with the molasses. Milk quality indicators, e.g., protein and casein content, remained unaffected after six months of the feeding trial. Similar to molasses, the Lba contributed to the improvement in lipid synthesis as 7.1% and 14.7% higher lipid content was observed in the milk of Group B and Group A compared with the initial values, respectively. The content of essential AAs such as isoleucine and valine was significantly higher in the experimental group fed with Lba rather than molasses. A similar trend of increase was found for branched-chain AAs, indicating an increase of 2.4% compared with the initial value. It was found that Group B tended to show higher PER1, PER2, and PER3 values after feeding with Lba than Group A fed with a high sucrose diet. The content of FAs in milk samples was affected by feeding. The highest content of MUFAs was observed in Group A, which received molasses. In contrast, the dietary inclusion of Lba promoted the increase in SFA and PUFA content in the milk after six months of the feeding trial.

To summarize, the use of Lba in the diet of dairy cows favorably affected the performance and quality traits of milk; however, to achieve better results, the optimization of feed intake with ingredients rich in carbohydrates and proteins such as molasses and biotechnologically produced Lba, respectively, can deliver the necessary energy levels for increased milk production and its quality.

## Figures and Tables

**Figure 1 animals-13-00815-f001:**
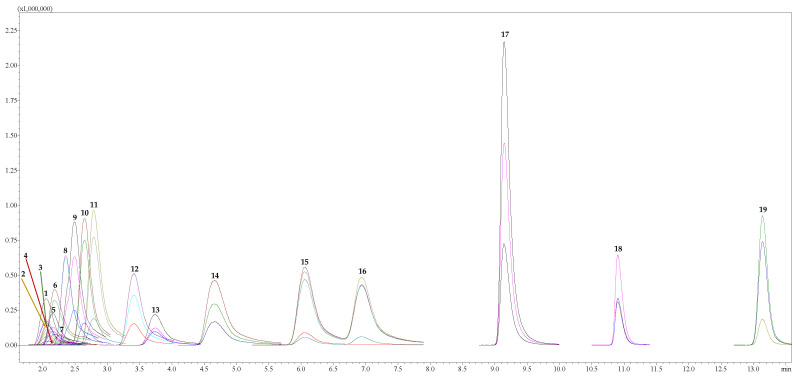
Selected ion chromatogram in MRM mode represents the profiles of 18 amino acids and one metabolite of tryptophan in a standard solution at the concentration of 2.5 μM L^−1^. ***Note:*** 1—Cystine (Cys); 2—Aspartic acid (Asp); 3—Serine (Ser); 4—Glycine (Gly); 5—Threonine (Thr); 6—Glutamic acid (Glu); 7—Alanine (Ala); 8—Proline (Pro); 9—Histidine (His); 10—Lysine (Lys); 11—Arginine (Arg); 12—Valine (Val); 13—Methionine (Met); 14—Tyrosine (Tyr); 15—Isoleucine (Ile); 16—Leucine (Leu); 17—Phenylalanine (Phe); 18—Tryptophan (Trp); and 19—Tryptamine (Trm).

**Table 1 animals-13-00815-t001:** Physical–chemical characteristics of prepared lactobionic-acid-rich whey used in the experiment as a supplement to the basic feed of lactating cows.

Quality Trait, %	Lba Rich Whey, 1 kg	Lba Rich Whey, 5 kg	Molasse, 1 kg **
Carbohydrates	15.1 ± 0.7 *	75.6 ± 3.7 *	45.5–62.1
Crude protein	3.7 ± 0.1	18.7 ± 0.4	6.0–13.5
Fat	0.06 ± 0.01	0.2 ± 0.0	0.2
pH	5.6 ± 0.1	5.6 ± 0.1	7.1
LBA, g L^−1^	11.3 ± 0.3	56.5 ± 4.3	–
Dry matter	22.5 ± 1.2	22.5 ± 1.2	75.1–84.0

***Note:*** The superscript single asterisk * indicates the content of lactose solely. The composition of molasses was retrieved from Palmonari et al., 2020, and Saric et al. [24,25] ** and expressed on a dry weight basis.

**Table 2 animals-13-00815-t002:** Physical–chemical characteristics of pre-concentrated commercially obtained mixed whey used as a carbon source for the production of lactobionic acid.

Indices, g L^−1^	Raw Whey
Total solids	20.7–25.1
pH	5.9–6.2
Lactose	15.3–18.0
Salts	2.2–2.5

**Table 3 animals-13-00815-t003:** Average lactating cow milk productivity, milk quality traits, and energy-corrected milk values between the study group and phase.

Quality Trait	Study Group			
	Group A		Group B	
	Study Phase			
	Begin	End	Begin	End
Milk yield, kg d^−1^	35.7 ± 2.9 ^aA^	28.8 ± 2.0 (<19.3) ^bA^	37.4 ± 2.0 ^aA^	28.8 ± 2.9 (<23.0) ^bA^
Fat content, %	3.4 ± 0.1 ^bB^	3.9 ± 0.2 (>14.7) ^aB^	4.2 ± 0.3 ^aA^	4.5 ± 0.4 (>7.1) ^aA^
Protein, %	3.3 ± 0.1 ^bB^	3.6 ± 0.1 (>9.1) ^aB^	3.8 ± 0.2 ^aA^	3.8 ± 0.2 (0)^a A^
Casein, %	2.6 ± 0.1 ^bB^	2.8 ± 0.2 (>7.7) ^aB^	3.0 ± 0.1 ^aA^	3.0 ± 0.1 (0) ^aA^
SCS	3.2 ± 0.8 ^bA^	3.5 ± 0.8 ^aA^	2.3 ± 0.4 ^aA^	3.0 ± 0.4 ^aA^
Urea, mg dL^−1^	23.3 ± 0.7 ^bA^	15.1 ± 0.9 (<35.2) ^aB^	23.5 ± 2.0 ^aA^	18.4 ± 0.9 (<21.7) ^bA^
ECM, kg d^−1^	32.5 ± 2.4 ^aB^	28.8 ± 2.0 (<11.4) ^bA^	39.7 ± 2.8 ^aA^	30.5 ± 2.2 (>37.8) ^bB^

***Note:*** Values are means ± SD of nine animals (*n* = 9). Different lowercase superscripts (^a,b^) in the same row indicate significant differences (Student’s *t*-test; *p* ≤ 0.05) between the study phase. Different uppercase (^A,B^) superscripts in the same row indicate significant differences (Student’s *t*-test; *p* ≤ 0.05) between study groups. Numbers in brackets indicate percentage increases (>) or decreases (<) in respective milk quality traits after the feeding trial. Group A—control group received molasses as a supplement to the feed; Group B—experimental group received biotechnologically obtained lactobionic acid as a supplement to the feed; SCS—somatic cell score; ECM—energy-corrected milk; and SD—standard deviation.

**Table 4 animals-13-00815-t004:** The profiles of amino acids in milk protein and their comparison between the group and study phase, g 100 g^−1^ DW.

Amino Acid	Study Group			
	Group A		Group B	
	Study Phase			
	Begin	End	Begin	End
	Flavor Amino Acids			
Alanine (Ala)	3.1 ± 0.0 ^aA^	3.2 ± 0.0 (>3.2) ^aA^	3.1 ± 0.0 ^aA^	3.3 ± 0.0 (>6.5) ^aA^
Arginine (Arg)	3.2 ± 0.0 ^aA^	3.2 ± 0.0 (0) ^aA^	3.3 ± 0.0 ^aA^	3.3 ± 0.0 (0) ^aA^
Asparagine (Asp)	7.7 ± 0.1 ^aA^	7.5 ± 0.2 (<2.6) ^bA^	7.4 ± 0.1 ^aB^	7.5 ± 0.0 (>1.4) ^aA^
Cysteine (Cys)	0.8 ± 0.0 ^aA^	0.8 ± 0.0 (0) ^aA^	0.8 ± 0.0 ^aA^	0.7 ± 0.0 (<12.5) ^aA^
Glycine (Gly)	1.9 ± 0.0 ^aA^	1.7 ± 0.0 (<10.5) ^bA^	1.9 ± 0.0 ^aA^	1.8 ± 0.0 (<5.3) ^aA^
Glutamine (Glu)	21.7 ± 0.2 ^aA^	21.3 ± 0.1 (<1.8) ^aA^	21.3 ± 0.1 ^aB^	20.7 ± 0.2 (<2.8) ^bB^
Proline (Pro)	9.2 ± 0.1 ^aA^	9.1 ± 0.0 (<1.1) ^aA^	9.1 ± 0.1 ^aA^	9.2 ± 0.1 (>1.1) ^aA^
Serine (Ser)	5.2 ± 0.0 ^aA^	5.3 ± 0.0 (>1.9) ^aA^	5.2 ± 0.0 ^aA^	5.3 ± 0.0 (>1.9) ^aA^
**∑_SUM_**	52.8 ± 0.4 ^aA^	52.1 ± 0.3 (<1.3) ^aA^	52.1 ± 0.4 ^aB^	51.8 ± 0.3 (<0.6) ^bB^
	**Essential Amino Acids**			
Histidine (His)	2.9 ± 0.0 ^aA^	3.0 ± 0.0 (>3.4) ^aA^	2.9 ± 0.0 ^aA^	2.8 ± 0.0 (<3.4) ^aA^
Isoleucine (Ile)	4.9 ± 0.0 ^bB^	5.1 ± 0.0 (>4.1) ^aB^	5.1 ± 0.0 ^bA^	5.4 ± 0.0 (>5.9) ^aA^
Leucine (Leu)	9.3 ± 0.1 ^aB^	9.1 ± 0.1 (<2.2) ^bB^	9.5 ± 0.1 ^aA^	9.5 ± 0.1 (0) ^aA^
Lysine (Lys)	8.2 ± 0.1 ^bB^	8.6 ± 0.0 (>4.9) ^aA^	8.4 ± 0.1 ^aA^	8.1 ± 0.0 (<3.6) ^bB^
Tyrosine (Tyr)	4.6 ± 0.0 ^bA^	4.9 ± 0.0 (>6.5) ^aA^	4.7 ± 0.0 ^bA^	4.9 ± 0.0 (>4.3) ^aA^
Threonine (Thr)	4.3 ± 0.0 ^aA^	4.2 ± 0.0 (<2.3) ^aA^	4.3 ± 0.0 ^aA^	4.1 ± 0.0 (<4.7) ^bA^
Valine (Val)	6.0 ± 0.0 ^aA^	5.9 ± 0.0 (<1.7) ^aB^	6.1 ± 0.0 ^aA^	6.3 ± 0.1 (>3.3) ^bA^
**∑_SUM_**	40.2 ± 0.2 ^aB^	40.8 ± 0.1 (>1.5) ^aB^	41.0 ± 0.2 ^aA^	41.1± 0.2 (>0.2) ^aA^
	**Flavor and Essential Amino Acids**			
Methionine (Met)	2.3 ± 0.0 ^aA^	2.4 ± 0.0 (>4.3) ^aA^	2.3 ± 0.0 ^aA^	2.3 ± 0.0 (0) ^aA^
Phenylalanine (Phe)	4.6 ± 0.0 ^aA^	4.7 ± 0.0 (>2.2) ^aA^	4.6 ± 0.0 ^aA^	4.7 ± 0.0 (>2.2) ^aA^
**∑_SUM_**	6.9 ± 0.0 ^aA^	7.0 ± 0.0 (>1.4) ^aA^	6.9 ± 0.0 ^aA^	7.0 ± 0.0 (>1.4) ^aA^
	**Branched-Chain Amino Acids**			
	20.2 ± 0.2 ^aB^	20.1 ± 0.1(<0.5) ^aA^	20.7 ± 0.2 ^bA^	21.2 ± 0.2 (>2.4) ^aA^
	**Amino Acids’ Quality Indices**			
**PER1**	3.1 ^aA^	3.0 (<3.2) ^aB^	3.2 ^aA^	3.2 (0) ^aA^
**PER2**	3.3 ^aA^	3.2 (<3.0) ^aA^	3.3 ^aA^	3.3 (0) ^aA^
**PER3**	2.7 ^aA^	2.4 (<11.1) ^bB^	2.7 ^aA^	2.6 (<3.7) ^aA^
**E/T, %**	44.9 ^bB^	45.5 (<1.3) ^aA^	45.6 ^aA^	45.8 (<0.4) ^aA^

***Note:*** Values are means ± SD of triplicates (*n* = 3) of an average milk sample of nine animals (*n* = 9). Different lowercase superscripts (^a,b^) in the same row indicate significant differences (Student’s *t*-test; *p* ≤ 0.05) between the study phase. Different uppercase (^A,B^) superscripts in the same row indicate significant differences (Student’s *t*-test; *p* ≤ 0.05) between the study groups. Numbers in brackets indicate percentage increases (>) or decreases (<) in respective milk quality traits after the feeding trial. Group A—control group received molasses as a supplement to the feed; Group B—experimental group received biotechnologically obtained lactobionic acid as a supplement to the feed; PER—protein efficiency ratio; E/T, %—the ratio of essential amino acids (E) to the total amino acids (T); DW—dry weight; and SD—standard deviation. Branched-chain amino acids are the sum of essential amino acids, i.e., leucine, isoleucine, and valine.

**Table 5 animals-13-00815-t005:** Fatty acid profile, cholesterol content, and estimated nutritional indexes of milk lipids in relation to the feeding of lactating dairy cows, % DW.

Fatty Acid	Abbreviation	Study Group			
		Group A		Group B	
		Study Phase			
		Begin	End	Begin	End
Undecanoic acid	C11:0	n.d.	n.d.	n.d.	n.d.
Dodecanoic acid	C12:0	6.4 ^aA^	5.5 (<14.1) ^bA^	6.0 ^aB^	5.5 (<8.3) ^bA^
Tridecanoic acid	C13:0	2.0 ^aA^	1.7 (<15.0) ^aA^	1.8 ^aA^	1.8 (0) ^aA^
Tetradecanoic acid	C14:0	11.8 ^bA^	12.8 (>8.5) ^aB^	11.8 ^bA^	12.9 (>9.3) ^aA^
Tetradecenoic acid	C14:1	2.5 ^aA^	2.3 (<8.0) ^aB^	2.4 ^aA^	2.4 (0) ^aA^
Pentadecanoic acid	C15:0	3.2 ^aA^	2.8 (<12.5) ^aA^	2.8 ^aB^	2.6 (<7.3) ^aB^
Pentadecenoic acid	C15:1	1.9 ^aA^	1.7 (<10.5) ^aA^	1.6 ^aB^	1.6(0) ^aA^
Hexadecanoic acid	C16:0	31.6 ^aB^	31.0 (<1.9) ^aB^	32.9 ^bA^	34.7 (>5.5) ^aA^
Heptadecanoic acid	C17:0	1.0 ^aA^	1.0 (0) ^aA^	0.9 ^aA^	0.9 (0) ^aA^
Heptadecenoic acid	C17:1	0.8	BLQ	BLQ	BLQ
Octadecanoic acid	C18:0	9.6 ^aA^	10.1 (>5.2) ^aA^	9.5 ^aA^	8.5 (<10.5) ^bA^
Octadecenoic acid	C18:1n9t	BLQ	0.8 (>100) ^A^	0.9 ^a^	0.8 (<11.1) ^bA^
Octadecenoic acid	C18:1n9c	18.4 ^bB^	20.5 (>11.4) ^aA^	20.9 ^aA^	19.4 (<7.2) ^bB^
Octadecadienoic acid	C18:2n6t	2.3 ^aA^	1.7 (<21.6) ^bA^	1.5 ^aB^	1.5 (0) ^aA^
Octadecadienoic acid	C18:2n6c	1.8 ^aA^	1.7 (<5.6) ^aA^	1.6 ^aA^	1.6 (0) ^aA^
Octadecatrienoic acid	C18:3n6c	0.7 ^a^	0.7 (0) ^aA^	BLQ	0.7 (>100) ^A^
Octadecatrienoic acid	C18:3n3c	0.8 ^aA^	0.6 (<25.0) ^aA^	0.7 ^aA^	0.7 (0) ^aA^
Eicosanoic acid	C20:0	0.4 ^a^	0.4 (0) ^aA^	BLQ	0.2 (>100) ^B^
CLA, Octadecadienoic acid	C18:2	0.7 ^aA^	0.5 (<28.6) ^aA^	0.6 ^aA^	0.6 (0) ^aA^
CLA, Octadecadienoic acid	C18:2	0.3 ^a^	0.3 (0) ^aA^	BLQ	0.3 (>100) ^A^
Eicosenoic acid	C20:1n9c	0.5 ^bB^	0.1 (<80.0) ^aB^	1.7 ^aA^	0.2 (<88.2) ^bA^
Heneicosanoic acid	C21:0	BLQ	BLQ	BLQ	0.4 (>100)
Eicosadienoic acid	C20:2n6c	BLQ	0.5 (>100)	BLQ	BLQ
Eicosatrienoic acid	C20:3n6c	BLQ	0.3 (>100)	BLQ	BLQ
Eicosatetraenoic acid	C20:4n6c	BLQ	0.3 (>100)	BLQ	BLQ
Eicosatrienoic acid	C20:3n3c	0.3	BLQ (<100)	BLQ	0.2 (>100)
Docosanoic acid	C22:0	2.1 ^aA^	1.5 (<21.6) ^bA^	1.4 ^bB^	1.7 (>21.4) ^aA^
Eicosapentaenoic acid	C20:5n3c	BLQ	0.4 (>100)	BLQ	BLQ
Docosadienoic acid	C22:2n6c	BLQ	BLQ	0.1	BLQ
Tetracosanoic acid	C24:0	0.9 ^aA^	0.4 (>55.6) ^bA^	0.6 ^aB^	0.3 (<50.0) ^bA^
Tetracosenoic acid	C24:1n9c	BLQ	0.6 (>100)	0.2	BLQ
Docosahexaenoic acid	C22:6n3c	BLQ	BLQ	BLQ	0.9 (>100)
**∑_SFAs_**		68.97 ^aA^	67.20 (<2.6) ^bB^	67.65 ^bB^	69.20 (>2.3) ^aA^
**∑_MUFAs_**		24.11 ^bB^	26.03 (>8.0) ^aA^	27.80 ^aA^	24.38 (<12.3) ^bB^
**∑_PUFAs_**		6.92 ^aA^	6.77 (<2.2) ^aA^	4.55 ^bB^	6.42 (>41.1) ^aA^
**CLA**		1.0 ^aA^	0.7 (<30.0) ^bB^	0.6 ^bB^	0.8 (>33.3) ^aA^
Cholesterol, mg 100g^−1^ DW		390.3 ± 27.1 ^bA^	421.9 ± 47.7 (>8.1) ^aA^	359.0 ± 12.1 ^bB^	394.8 ± 1.8 (>10.0) ^aB^
**PUFA/SFA**		0.1 ^aA^	0.1 (0) ^aA^	0.1 ^aA^	0.1 (0) ^aA^
**IA**		2.8 ^bA^	3.6 (>28.6) ^aA^	2.4 ^bB^	3.4 (>41.7) ^aB^
**IT**		3.2 ^aA^	3.0 (<6.3) ^aB^	3.0 ^bB^	3.5 (>16.7) ^aA^
**HH**		0.5 ^aA^	0.5 (0) ^aA^	0.5 ^aA^	0.5 (0) ^aA^
**HPI**		0.4 ^aA^	0.4 (0) ^aA^	0.4 ^aA^	0.3 (<25.0) ^aA^

***Note*:** Values are means ± SD of duplicates (*n* = 2). Different lowercase superscripts (^a,b^) in the same row indicate significant differences (Student’s *t*-test; *p* ≤ 0.05) between study phases. Different uppercase (^A,B^) superscripts in the same row indicate significant differences (Student’s *t*-test; *p* ≤ 0.05) between study groups. SFA—saturated fatty acids; MUFA—monounsaturated fatty acids; PUFA—polyunsaturated fatty acids; CLA—conjugated linoleic acid; IA—index of atherogenicity; IT—index of thrombogenicity; HH—ratio of hypocholesterolemic to hypercholesterolemic levels; HPI—health-promoting index; BLQ—below limit of quantification; DW—dry weight; SD—standard deviation; and n.d.—not detected.

## Data Availability

Data are available from authors upon reasonable request.

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
