# Peer review of "The Impact of Biotechnologically Produced Lactobionic Acid in the Diet of Lactating Dairy Cows on Their Performance and Quality Traits of Milk"

_animals, 2023, doi:10.3390/ani13050815_

Round 1

Reviewer 1 Report

Dear Editor

Thank you for giving me the opportunity to evaluate this manuscript entitled “The Impact of Biotechnologically Produced Lactobionic Acid in the Diet of Lactating Dairy Cows on their Performance and Quality Traits of Milk”. Optimum use of waste, despite the fact that it will reduce the feed cost, prevents their release in the environment. The manuscript was properly conducted and findings reported are important for Animals. The paper contains valuable data. For further improvement of the manuscript, it requires use newly published manuscripts in this field. And author should address the work novelty.

You can use new references such as:

Sharifi, M., Taghizadeh, A., Hosseinkhani, A., Mohammadzadeh, H., Palangi, V., Macit, M., ... & Abachi, S. (2022). Nitrate supplementation at two forage levels in dairy cows feeding: milk production and composition, fatty acid profiles, blood metabolites, ruminal fermentation, and hydrogen sink. Anim. Sci, 22(2).

Kindly regards

Author Response

Authors: Dear reviewer. The authors would like to thank the reviewer for carefully checking our manuscript and for your valuable comments. In preparing the manuscript authors have incorporated most of the changes suggested. The authors refer to them in detail below.

Reviewer: Thank you for giving me the opportunity to evaluate this manuscript entitled “The Impact of Biotechnologically Produced Lactobionic Acid in the Diet of Lactating Dairy Cows on their Performance and Quality Traits of Milk”. Optimum use of waste, despite the fact that it will reduce the feed cost, prevents their release in the environment. The manuscript was properly conducted and findings reported are important for Animals. The paper contains valuable data. For further improvement of the manuscript, it requires use newly published manuscripts in this field. And author should address the work novelty.

You can use new references such as:

Sharifi, M., Taghizadeh, A., Hosseinkhani, A., Mohammadzadeh, H., Palangi, V., Macit, M., ... & Abachi, S. (2022). Nitrate supplementation at two forage levels in dairy cows feeding: milk production and composition, fatty acid profiles, blood metabolites, ruminal fermentation, and hydrogen sink. Anim. Sci, 22(2).

Authors: The authors appreciate the reviewer's suggestion to include an additional reference in the manuscript very much. The authors have supplemented the manuscript with additional information supported by doi:10.2478/aoas-2021-0044.

On behalf of all the co-authors

Yours sincerely,

Vitalijs Radenkovs

Principal investigator, Division of Smart Technologies, Research Laboratory of Biotechnology, Latvia University of Life Sciences and Technologies, LV-3004 Jelgava, Latvia 

Reviewer 2 Report

The manuscript appears interesting and comprehensive. I consider implementing the statistical part with one-way and two-way ANOVA analysis for fatty acids in order to evaluate variation of each fatty acid pre and post and between control and treated. Graphpad can be used for the complete statistical analysis.

Author Response

Authors: Dear reviewer. The authors would like to thank the reviewer for carefully checking our manuscript and for your valuable comments. In preparing the manuscript authors have incorporated most of the changes suggested. The authors refer to them in detail below.

Reviewer: The manuscript appears interesting and comprehensive. I consider implementing the statistical part with one-way and two-way ANOVA analysis for fatty acids in order to evaluate variation of each fatty acid pre and post and between control and treated. Graphpad can be used for the complete statistical analysis.

Authors: The authors appreciate the reviewer's suggestion to include statistical analysis for fatty acids to make the obtained results more representative. Additional statistical analysis by one-way ANOVA has been done.

On behalf of all the co-authors

Yours sincerely,

Vitalijs Radenkovs

Principal investigator, Division of Smart Technologies, Research Laboratory of Biotechnology, Latvia University of Life Sciences and Technologies, LV-3004 Jelgava, Latvia 

Reviewer 3 Report

Make changes according to the suggestions 

Author Response

Authors: Dear reviewer. The authors would like to thank you for carefully checking our manuscript and valuable comments. In preparing the manuscript authors have incorporated most of the changes suggested. The authors refer to them in detail below.

Reviewer: Replace EXP with some group name like Group A and Group B as control group, its look inappropriate EXP and CON

Authors: The authors are grateful to the reviewer for the reasonable suggestion. Now in the manuscript Group A and Group B refer to the control and experimental group, respectively.

Reviewer: Rephrase line 66 drying whey  ???????

Authors: Rephrased. Thank you.

Reviewer: Replace is with are

Authors: Replaced. Thank you.

Reviewer: Including dont use inclu?????

Authors: The authors considered the reviewer’s suggestion. Much appreciated.

Reviewer: As for as the chemical composition of whey is discussed it varies.......

Authors: The proposed sentence has been included. Thank you.

Reviewer: included in the current study

Authors: Added. Thank you.

Reviewer: replace was with were

Authors: Dear reviewer. The authors are not native in English but will try to explain. In the sentence “The basic feed for both groups was prepared directly….” The subject in this sentence is “the feed, " a singular form. So more correct to use the singular past tense “was”.

Reviewer: There is no need to elaborate this part, make it more precise????

Authors: The authors try to stick to one style in their works, indicating the purity and source of chemicals and standards used in the study. This information allows readers to reproduce the experiment. The authors shortened this section as much as possible. Thank you.

Reviewer: supplements

Authors: Corrected.

Reviewer: they are same?????

Authors: Dear reviewer. The increase in fat content of the CON group amounted to 14.7%, while in EXP 7.1% compared with the initial values.

Reviewer: there should be uniformity in writing SCC or SCS so Use SCC instead of SCS

Authors: The authors apologize for misleading the reviewer with the terms “SCC” and “SCS”. However, both of them must be apparent in the manuscript since the authors converted standardized values of SSC (Somatic cells count) to SCS (somatic cell score) for a better understanding of the results. The authors hope that this ensures additional clarity on this matter.

Reviewer: indices is more appropriate for research

Authors: The authors considered the reviewer’s suggestion. Much appreciated.

Reviewer: conclusion needs to more summarization of the text, make it more precise and comprehensive but short text.

Authors: The authors have revised the conclusion part. Thank you.

Authors: The manuscript has been proofread to exclude grammar and typos mistakes.

Authors: Dear reviewer. The authors would like to thank you for carefully checking our manuscript and valuable comments.

On behalf of all the co-authors

Yours sincerely,

Vitalijs Radenkovs

Principal investigator, Division of Smart Technologies, Research Laboratory of Biotechnology, Latvia University of Life Sciences and Technologies, LV-3004 Jelgava, Latvia 

Round 2

Reviewer 3 Report

the changes are according to the suggestions so now accept this article.
